



# 1 Characterizing Urban Planetary Boundary Layer
# 2 Dynamics Using 3-Year Doppler Wind Lidar
# 3 Measurements in a Western Yangtze River Delta City,
# 4 China

Tianwen Wei[1], Mengya Wang[1*], Kenan Wu[1], Jinlong Yuan[1], Haiyun Xia[1,2*], Simone
Lolli[3]
[1]School of Atmospheric Physics, Nanjing University of Information Science & Technology, Nanjing
210044, China.
[2]School of Earth and Space Science, University of Science and Technology of China, Hefei 230026,
China
[3]CNR-IMAA, Contrada S. Loja snc, Tito Scalo (PZ), 85050, Italy
*Correspondence to*: Mengya Wang (wmengya123@nuist.edu.cn) and Haiyun Xia (hsia@ustc.edu.cn)

## 13 Abstract

Understanding the dynamics of the planetary boundary layer (PBL) is crucial for comprehending
land-atmosphere interactions. This study utilizes three years of Doppler wind lidar measurements from
June 2019 to June 2022 to investigate PBL dynamics over Hefei, a city in the Western Yangtze River
Delta, China. We focus on the seasonal and diurnal variations in key characteristics, such as wind profiles,
shear intensity, turbulent mixing, low-level jets (LLJs), and mixing layer heights (MLH). Results show
that horizontal wind speeds accelerated more rapidly above 3 km, with the predominant westerly winds
(270°±15°) in all seasons. The vertical depth of high wind zone (> 8 m s-1) during the day is found
generally deeper than at night, particularly in winter. In Hefei, LLJs primarily form at sunset and dissipate
by noon, typically at altitudes between 0.5 and 0.6 km throughout the year, except in July. LLJ
occurrences are most frequent in spring (31.7%), followed by summer (24.7%), autumn (22.3%), and
winter (21.3%). Summer LLJs are most intensified, extending up to 1.5 km. The larger wind gradient
below the jets significantly enhances turbulence and shear intensity near the ground at night. The seasonal
average MLH peaks between 2:00 p.m. and 3:00 p.m., reaching approximately 1.2 km in spring and
summer. Cloud cover raises MLH by about 100 m at night but decreases it by 200 m at the afternoon
peak. This study provides insights into lidar-based PBL dynamics and highlights implications for local
standards concerning low-altitude economic activities.

## 30 1. Introduction

The planetary boundary layer (PBL) refers to the lowest 1~3 km of the atmosphere that is directly
influenced by the presence of the underlying surface, and usually responds to surface forcings in an hour
or less (Stull, 1988). These surface forcings include frictional drag, heat transfer, pollutant emission,
evaporation and transpiration, and terrain induced flow modifications (Garratt, 1994). The depth and
structure of the PBL are determined by the physical and thermal properties of the underlying surface as
well as the dynamics and thermodynamics of the lower atmosphere (Madala et al., 2014). One of the
most important characteristic of the PBL is turbulence, which dominates the vertical exchange of heat,





moisture, momentum, trace gases, and aerosols between the free atmosphere and the Earth's surface or
regolith (Baklanov et al., 2011; Petrosyan et al., 2011). In the PBL, the sources of turbulent mixing exhibit
significant temporal and spatial variations, which include buoyancy (convective mixing), wind shear
(mechanical mixing), entrainment at the top of boundary layer, and radiative cooling in stratocumulus
clouds (top-down convective mixing) (Ortiz-Amezcua et al., 2022). Such turbulent motion in the PBL
has been demonstrated to be inherently connected to air pollution by modulating the dispersion, transport,
and accumulation process, and have critical impacts on land-atmosphere energy balance, as well as
aerosol-cloud-precipitation interactions (Kim and Entekhabi, 1998; Wang et al., 2001; Chen et al., 2011;
Wood et al., 2015; Li et al., 2017; Su et al., 2020, 2018; Christensen et al., 2024).
47        Hefei, the capital of Anhui province, has experienced incredible economic growth and urban sprawl
over the past two decades (Zhao and Zou, 2018). Situated between the Yangtze River and Huaihe River,
in what is known as the Jianghuai region, the Hefei Metropolitan Circle plays a pivotal role in the Yangtze
River-Huaihe River Water Transfer Project to provide benefits for water supply, transportation,
agriculture, and power generation (Li et al., 2019; Zhang et al., 2023). Apart from tremendous economic
benefits achieved in Hefei, intense human activities create a profound influence on the local climate,
affecting the thermal, hydrological, and wind environments in the PBL within and beyond city limits (Shi
et al., 2008; Li et al., 2022a). In this context, the PBL study is vital for better understanding the exchange
process between the atmosphere and land over complex underlying surfaces, and improving the
parameterization schemes in numerical weather prediction models. However, previous studies mainly
focused on surface air pollution characteristics and its associations with meteorological parameters, as
well as the impacts on human health based on in-situ monitoring measurements or air quality modelling
(Hu et al., 2024; Qin et al., 2017; Shen et al., 2022; Zhang et al., 2017; Zhu et al., 2019). Among various
observation techniques, the lidar is a powerful tool and has been applied in retrieve vertical profiles of
PBL properties in Hefei, such as aerosols, winds, turbulence, precipitation, temperature, and water vapor
during a period (Zhou, 2002; Xia et al., 2015, 2016; Wei et al., 2021, 2022; Jiang et al., 2022; Yuan et al.,
2020; Wang et al., 2015b). Therefore, it is essential to utilize the long-term lidar measurement to
characterize the PBL dynamics such as winds and turbulence sources to further understand the land-
atmosphere interaction.
66        The key parameter of PBL meteorology is the PBL height (PBLH) which displays significant
spatiotemporal variability under different atmospheric and surface conditions (Guo et al., 2019; Zhang
et al., 2022; Zhao et al., 2023). It strongly depends on surface characteristics such as surface heating rate,
strength of winds, topography, surface roughness, free atmospheric characteristics, the amount of clouds
and moisture (Kotthaus et al., 2023; Zhang et al., 2020). Multiple approaches have been developed to
determine the PBLH based on observations, such as in situ radiosonde (Gu et al., 2022; Guo et al., 2021;
Yue et al., 2021), aerosol-based and dynamic-based lidar techniques (Chen et al., 2022; Huang et al.,
2017; Vivone et al., 2021; Wang et al., 2020, 2021; Yang et al., 2020; Yin et al., 2019). In the practical
measurements of PBLH, it is necessary to consider its distinct diurnal cycle of PBL. The PBL can be
categorized into three dominant regimes: convective boundary layer (CBL), stable boundary layer (SBL),
and residual layer (RL) based on the thermodynamic stability in the lower atmosphere (Caughey and
Palmer, 1979). After sunrise, increasing radiative heating triggers the development of near-surface
turbulent eddies and leads to the formation of CBL, which the CBL grows with time and reaches its
maxima in the early afternoon. The CBL consists of a convective surface layer, mixing layer (ML) above,
and entrainment zone (EZ) at the top (Wyngaard, 1988). After sunset, the radiative cooling creates the
SBL close to the surface and its depth grows as night progresses. The RL lies above the SBL meanwhile


a capping inversion overlies the RL (Fochesatto et al., 2001). However, studies in diurnal and seasonal
characteristics of the PBLH under different stable conditions in Hefei based on long-term measurements
have not been documented yet, to the best of our knowledge at the writing of this work.
Turbulence in the PBL is generated mechanically by wind shear, and convectively by buoyancy.
Wind shear is the main source of turbulence in the nocturnal boundary layer (NBL, also known as the
SBL), which can be enhanced in the presence of low-level jets (LLJs). Yang et al. (2023) found that wind
shears induced by LLJs often enhanced the vertical mixing processes, reduced the atmospheric stability,
and resulted in small weak direction shifts in eastern Idaho, USA. The formation of LLJ can provide a
driving force for the development of a deeper CBL on the Tibet Plateau (Su et al., 2024). Many studies
investigated the prominent role of LLJs in heavy rainfall events in the Jianghuai region (Chen et al., 2020;
Yan et al., 2021; Liu et al., 2022; Cui et al., 2023), but there has been a lack of research specifically
focusing on Hefei. The Huaihe River region, including Hefei, is one of the six high-frequency regions of
LLJs in China (Yan et al., 2021). The LLJs over China are usually classified into two types: boundary
layer jets (BLJs, below 1 km) and synoptic-system-related LLJs (SLLJs, within 1–4 km) (Du et al., 2014).
The occurrence of BLJs is associated with significant vertical shear of horizontal wind and diurnal
variation. On the contrary, SLLJs are usually related to synoptic-scale weather systems. This study
addresses a previous research gap by investigating the characteristics of LLJs formation and types, and
vertical wind shear (VWSH) in Hefei, with a focus on their monthly variations across different times and
altitudes.
In this paper, we utilize a 3-year Doppler wind lidar measurements to characterize the PBL
dynamics in Hefei. The horizontal wind speeds and direction, LLJs, VWSH, turbulent kinetic energy
dissipation rate (TKEDR), mixing layer height (MLH) and PBLH are thoroughly analyzed. Remote
sensing retrieval of the above PBL parameters have been fully illustrated and validated in our previous
studies (Wang et al., 2019, 2021; Wei et al., 2019, 2022; Wang et al., 2024). This paper aims to shed new
light on the diurnal and seasonal characteristics of PBL meteorology and turbulence influenced by diurnal
cycles, general circulation, the Asian monsoon, and the synoptic systems.

## 2. Materials and methodology

### 2.1 Study area and instruments

Hefei, a rapidly developing new first-tier city, is located in Eastern China within central Anhui
Province in Figure 1(a). It covers an area of 11465 km$^2$, comprising four urban districts, one county-level
Chaohu city, and four counties. Its topography includes flat plains, gently rolling hills, and major water
bodies such as Chaohu Lake to the southeast in Figure 1(b). The city altitude mainly ranges from 15 to
81 m, with the highest point reaching 595 m (Sun and Ongsomwang, 2021). The Dabie Mountain in the
southwest introduces varied elevations and complex topographical features that influence regional
atmospheric dynamics in Hefei. Anhui province including Hefei, is located across both the eastern
monsoon region and the north-south climate transition zone of China. Hence, Hefei is characterized by
the typical subtropical monsoon climate with four distinct seasons. The city receives an annual
precipitation of ~1000 mm and average temperature of 15.7 °C, with prevailing southeast winds in spring
and summer and northwest winds in autumn and winter (Li et al., 2024).

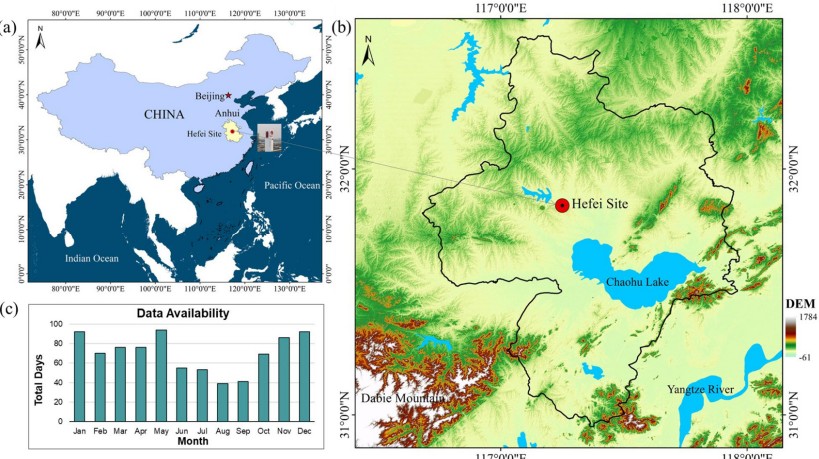

**Figure 1.** Study area and location of the Doppler wind lidar system. (a) Location (31.83°N, 117.25°E) of Hefei site and administrative boundary of Anhui province; (b) DEM, and the solid black line represents the administrative boundary of Hefei city; (c) Data availability of 3-year Doppler wind lidar measurements. Total days with valid lidar measurements are accounted for each month.

A compact coherent Doppler wind lidar (CDWL) system was deployed on the roof of the School of Earth and Space Science (SESS) building of the University of Science and Technology of China (31.83° N, 117.25°E) in the urban area of Hefei, to monitor the vertical profiles of aerosol, cloud and wind field. The specific location of lidar is referred as Hefei site in Figure 1(a) and Figure 1(b). The lidar system operates at 1.5 µm eye-safe wavelength and uses 300 µJ pulse energy and 10 kHz repetition rate to achieve a maximum detection range of up to 15 km. During the long-term experiment, the lidar performed continuous velocity azimuth display (VAD) scanning mode for high spatial-temporal resolution wind profile measurement. The azimuth angle ranges from 0° to 300° with an interval of 5° and the elevation angle is 60°. The key parameters of the Doppler lidar system are summarized in Table A1 in Appendix. Detailed information about the validation and application of the lidar system can be found in our previous works (Jia et al., 2019; Wei et al., 2020, 2021). The data availability is presented in Figure 1(c) with monthly statistics of total valid days. Note that the lower data availability during the summer seasons is primarily due to frequent rainfall and high temperatures, which caused instability in the lidar systems. However, these issues have been significantly improved in the recently updated systems (Xia et al., 2024).

## 2.2 Datasets and methods

The CDWL system operated for three consecutive years from June, 2019 to June, 2022, except for some maintenance interruptions (Wang et al., 2024). The number of days available for different seasons and weather types is presented in Table 1, respectively.

**Table 1.** The days of different weather types during the period of Doppler lidar operation

| Weather Types[*] | Spring | Summer | Autumn | Winter | Total days |
|---|---|---|---|---|---|
| Rainy | 62 | 44 | 37 | 75 | 218 |
| Clear | 69 | 21 | 47 | 76 | 213 |
| Cloudy | 68 | 82 | 50 | 64 | 264 |

| | | | | | |
|---|---|---|---|---|---|
| Partly Cloudy | 39 | 38 | 44 | 39 | 160 |

*Rainy: rain persists for more than 2 hours **Clear:** clouds are present for less than 2 hours.
**Cloudy:** cloud coverage exceeds 8 hours. **Partly Cloudy:** cloud coverage lasts between 2 to 8 hours.

The time resolution and range gate resolution of the original radial measurements are 1 s and 30 m, respectively. Horizontal wind speed (HWS), horizontal wind direction (HWD), and vertical wind speed (VWS) are retrieved from the measured radial speeds at different azimuth angles using a filtered sin-wave fitting method, based on the assumption of horizontally homogenous wind field (Smalikho, 2003; Banakh et al., 2010; Wei et al., 2020). Considering the duration of one VAD scan, the time resolution of wind profile becomes about 2 minutes. Here, the wind direction of 0° represents the horizontal wind coming from the north, and the angle increases clockwise. The negative (positive) vertical wind speed was defined as upward (downward) motion in this study.

Turbulence activity can be expressed by vertical velocity variance, spectrum width, turbulent kinetic energy, and TKEDR (O'Connor et al., 2010). In this study, we estimate TKEDR using the turbulence statistical model based on the relation between the structure-function of the measured radial velocity and theoretical value (Banakh et al., 2017). The MLH is a significant parameter for presenting the vertical turbulent exchange within the PBL. On the basis of the characteristics of decreasing convective turbulence intensity along with height, the threshold method can effectively determine a typical turbulence height. Here, the MLH is defined as the height up to which TKEDR > $10^{-4}$ m$^2$ s$^{-3}$ is reached (Banakh et al., 2021; Wang et al., 2021). In addition, the aerosol-based PBLH, shown in Section 3.6, is also calculated for comparison. It is determined from the aerosol backscatter coefficient using a Harr wavelet method (Caicedo et al., 2017; Kotthaus et al., 2023).

LLJ is a fast air stream with a wind speed maximum in the lowest kilometers of the troposphere (Stull, 1988). Referring the previous studies (Qiu et al., 2023; Zhang et al., 2018; Tuononen et al., 2017) and considering the local characteristics, we identify the LLJs at Hefei using the following criteria: (1) the maximum wind speed $U_{max} >$ 8 m $s^{-1}$and, (2) the wind speed difference $\Delta U = U_{max} - U_{min} >$ 2.6 m $s^{-1}$, where $U_{min}$ is the minimum wind speed above the height of $U_{max}$. The LLJ height is then defined as the height of $U_{max}$. In addition, when a two-layer LLJ exists, the lower one will be selected. Each wind profile was applied to identify the LLJ event. In the statistics procedure of Section 3.3, a time window of 1 h is used to filter out the outliers, and those with fewer than 60% within the window were abandoned.

Vertical wind shear (VWSH) is defined as the change in wind speed and/or direction with height. It can be calculated from the vertical wind profiles using the following equation (Manninen et al., 2018)

$$\mathrm{VWSH} = \frac{(\Delta u^2 + \Delta v^2)^{0.5}}{\Delta z}$$

where the difference in vectors of the wind components $u$ and $v$ is divided by the height difference $\Delta z$ between the two altitude levels used to compute the wind shear.



# 3. Results

## 3.1 The 3-year seasonal profiles of the wind frequency

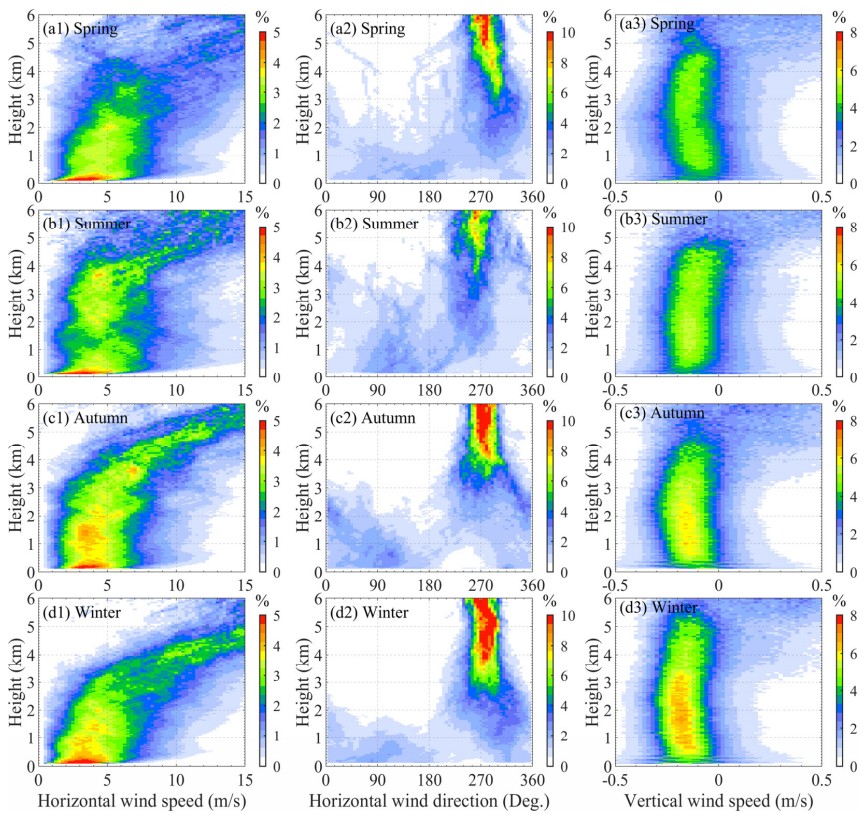

**Figure 2.** The seasonal frequency distributions of horizontal wind speed (left panel), horizontal wind direction (middle panel), and vertical wind speed (right panel) at different heights below 6 km during (a) Spring: Mar-May; (b) Summer: Jun-Aug; (c) Autumn: Sep-Nov; and (d) Winter: Dec-Feb, at Hefei. Note that the sum of all frequency values along the x-axis equals 100% at any specific height. It should be noted that negative value of vertical wind speed is defined as the upward movement of air.

Vertical wind profiles are influenced by surface friction, terrain, local pressure systems, and global atmospheric circulation patterns. We retrieve the vertical profiles of HWS, HWD, and VWS and calculate the frequency (%) of their occurrence at different heights above ground level (AGL), as shown in Figure 2. The frequency distribution is calculated by the ratio of the counts of wind speeds falling into each bin on the x-axis to the total valid numbers at each height. Therefore, the sum of all frequency values along the x-axis is 100% at any specific height. To represent rich details, the bin size or resolution (i.e., the width of each column) is set to 0.25 m s$^{-1}$, 5°, and 0.02 m s$^{-1}$ for HWS, HWD, and VWS, respectively.

In the left panel of Figure 2, the frequency distribution of HWS (hereafter referred to as HWS%) exhibits a rightward skew in all seasons, a characteristic often modeled using a Weibull or Lognormal distribution due to the non-negative nature of wind speed (Justus et al., 1978; Pobočíková et al., 2017).





Close to the ground, the majority of HWS values are clustered at the lower end, mainly as a result of
surface friction. Below ~300 m AGL, HWS increase rapidly as surface friction decreases. From 300 m
to 3 km AGL, HWS increases steadily while becoming more dispersed, with the overall distribution
(HWS%) spanning between 2 and 7 m s$^{-1}$. Above 3 km, HWS accelerates more rapidly, particularly in
autumn and winter, where HWS% remains relatively concentrated. In contrast, HWS% in spring and
summer is more dispersed with a lower frequency of high HWS occurrences (> 10 m s$^{-1}$). Many studies
have demonstrated a significant decrease trend of near surface wind speed in eastern China including
Anhui province, induced by large-scale circulation and local land use and land cover change (Li et al.,
2018; Liu et al., 2023; Li et al., 2022). Wang et al. (2015) observed that the value of annual mean surface
wind speed in Hefei city during 1981-2012 was between 2.0 m s$^{-1}$ and 2.6 m s$^{-1}$ and the highest frequency
of maximum surface wind speed occurred in spring. A recent study by Li et al. 2022a analyzed the
maximum daily wind speed of 10 minutes from 51 meteorological stations in Anhui province from 2006
to 2020, which showed that the average maximum wind speed in the city of Hefei was between 9.1~17.6
m s$^{-1}$. Therefore, our results of seasonal HWS values near the ground correspond to previous studies.

212        The frequency distribution of HWD (hereafter referred to as HWD%) exhibits distinct vertical
characteristics, as shown in the middle panel of Figure 2. At higher altitudes (> 3 km), the distribution of
HWD is much more concentrated, with predominant westerly winds (270°±15°) in all seasons. Because
Hefei city is located between 31°4′ N and 32°38′ N, which is affected by westerly circulation. The finding
of prevailing westerlies throughout the year in Hefei is consistent with (Sun et al., 2021). In contrast, the
influence of westerlies on HWD% below the PBL is insignificant due to the impact of the underlying
surface roughness, terrain distribution, and air flow turbulence. Below 3 km AGL, we can discover
notable southwest winds in summer compared to the other seasons. In the summer monsoon season,
eastern China (including Hefei city) is mainly dominated by southwest winds, as has been reported by
many studies (Liu et al., 2015; Yan et al., 2022; Zhao et al., 2007). Wind directions in the PBL tend to be
more variable and chaotic compared to those at higher altitudes. And westerly winds above 1.5 km
consistently strengthen with increasing altitude in all seasons.

224        The right panel of Figure 2 illustrates seasonal profiles of VWS frequency (hereafter referred to as
VWS%). The frequency distribution of VWS% is right-skewed and its center lay in negative values
between -0.2 m s$^{-1}$ and -0.1 m s$^{-1}$. The results show that most VWS values are negative below 5 km in all
seasons, representing upward motion in the atmosphere. It demonstrates the asymmetric nature of vertical
velocities in the atmosphere, where upward movements are stronger than downward movements
(Tamarin-Brodsky and Hadas, 2019). Furthermore, Figure 2 (d3) shows that winter has the highest
frequency of negative VWS, with most VWS% ranging from 4% to 7% below 3 km AGL. A climatology
study of cold frequency suggests that cold fronts are most frequently occurred in cold seasons over Hefei
city (Xue et al., 2022). In winter, cold fronts associated with the winter monsoon can enhance upward
motion of the air as the heavier (more dense) cool air pushes under the lighter (less dense) warm air
(Kang et al., 2019; Parsons, 1992). The upward motion intensifies and is vigorous along the frontal
boundaries, leading to cloud formation and precipitation. The higher positive values in the asymmetric
distribution of VWS, particularly above 3 km, are attributed to the contribution of falling precipitation
particles (Wei et al., 2019). Under these conditions, the detected vertical speed reflects the movement of
larger hydrometeors rather than the air motion itself.

## 3.2 Diurnal HWS profiles in different seasons

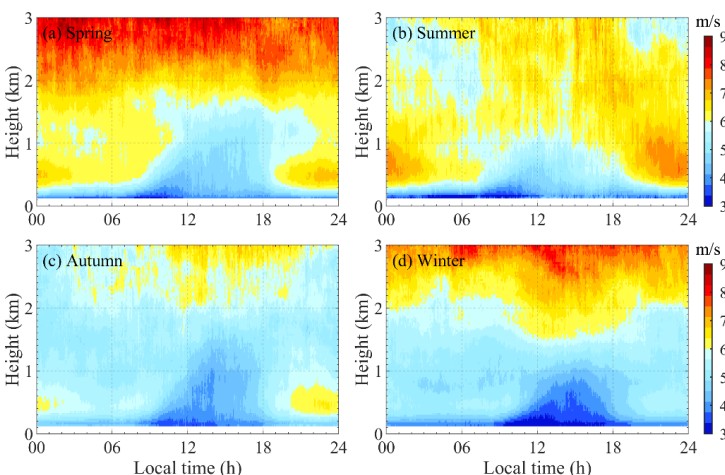

**Figure 3.** Diurnal profiles of seasonal averaged horizontal wind speeds at different heights below 3 km during (a)
Spring: Mar-May; (b) Summer: Jun-Aug; (c) Autumn: Sep-Nov; and (d) Winter: Dec-Feb, at Hefei.

The diurnal variation of the vertical wind profile within the PBL is intricately linked to the dynamics
and thermodynamics driven by the daily cycle of solar heating and longwave cooling. Figure 3 illustrates
how HWS profile varies with local time (LT) on a seasonal scale. The minimum values of HWS are
found in the lowest layer, primarily due to the impact of rough surface.

During the day, solar heating induces turbulence and convection, which increase surface friction
and slow down the up-level horizontal wind. This results in the formation of a gentle wind zone (GWZ),
characterized by wind speeds below 5 m s$^{-1}$, a feature that can be observed in all seasons. And the diurnal
variation of the GWZ strongly correlates with the development of the mixing layer. At night, radiative
cooling generates a temperature inversion, inhibiting vertical mixing and fostering laminar flow with
increased shear intensity. Consequently, nocturnal winds are generally stronger than daytime winds at
the same height below 1.5 km AGL throughout all seasons. Above this height, the HWS profile is usually
more uniform and stronger due to the reduced frictional drag in the free atmosphere. It is interesting to
find that the vertical height of high wind zone (> 8 m s$^{-1}$) during the day is much lower than at night,
particularly in winter. In Figure 3d, an appreciable enhancement of HWS at 1.5 km is discovered during
the day particularly between 11:00 a.m. and 16:00 p.m., when the PBL tends to grow and become deeper
due to radiative heating of the surface. In general, the HWS increases with height. However, as seen in
Figures 3(a) to 3(c), a distinct local maximum in HWS, occurring between approximately 0.4 km and 0.8
km, is observed after 8:00 p.m. and before 7:00 a.m. the next day. This is especially pronounced in
summer, where the highest values and the highest vertical extent of the wind are recorded. These winds
are typically associated with the nocturnal LLJs, a narrow band of strong winds that forms in the lower
PBL. Although the seasonal average HWS reflects the overall wind conditions, the pronounced notch
structure in the profile underscores the frequent occurrence of LLJs, which will be explored in more
detail in the next section.

## 3.3 Monthly characteristics of LLJ at different times and heights

LLJ is characterized by a concentrated band of strong winds located in the lower part of the



268 atmosphere. The diurnal variation of its formation and occurrence is influenced by the interaction
269 between surface heating/cooling cycles, atmospheric stability, and synoptic-scale weather patterns.
270 Figure 4(a) illustrates the statistical frequency (%) of the occurrences of LLJs at different hours for each
271 month. Frequency values are calculated as the ratio of the total number of LLJ occurrences to the total
272 number of available days in the specific month over a 3-year period. Additionally, the monthly variation
273 of the sunrise, noon, and sunset time was also plotted. Figure 4(b) presents the frequency distribution (%)
274 of LLJs occurrences over the height for each month, with the sum of each column equaling 100%. The
275 wind rose charts of the LLJ events for the four seasons are presented in Figure 5(a)~(d), respectively.
276 The seasonal and intraseasonal variability of predominant wind directions and wind speeds of LLJs are
277 influenced by general circulation, the East Asian monsoon, and synoptic systems. The spatial
278 distributions of the 500-hPa geopotential height and geopotential height anomalies are presented in
279 Figure A1.

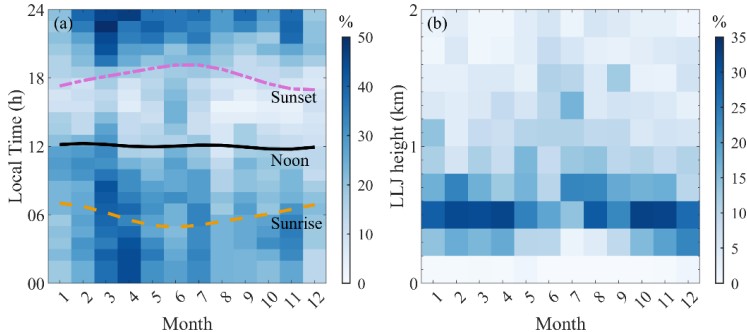

281 **Figure 4.** (a) The frequency (%) of LLJs occurrences at different times for each month. The purple dot-dashed line,
282 the black solid line, and the orange long-dashed line refer to the mean sunset, noon, and sunrise times for each month,
283 respectively. (b) The frequency distribution of LLJ occurrence over the height for each month, with the sum of each
284 column equal to 100%.

285 Generally, LLJs occurrences are most frequent in spring, followed by summer, autumn, and winter
286 in Hefei. The average seasonal LLJs frequencies were 31.7%, 24.7%, 22.3%, and 21.3% in spring,
287 summer, autumn, and winter, respectively. In Figure 4(a), the sunrise and sunset times exhibit monthly
288 variation due to the Earth's revolution around the Sun. We refer to the period between sunset and sunrise
289 as daytime, and the period between sunset and the next sunrise as nighttime. LLJs are more frequently
290 observed during the night and early morning throughout all months. In the classical theoretical
291 description of inertial oscillations, LLJs develop because of the decoupling of nocturnal winds from the
292 surface friction, facilitated by the formation of a near-surface temperature inversion (Blackadar, 1957).
293 At night, the surface cools more rapidly than the air above, giving a rise to the formation of temperature
294 inversions. It causes the air above the temperature inversion to decouple from the surface's frictional
295 effects (Mirza et al., 2024). The weaker friction enables an acceleration of wind aloft with the
296 development of a pronounced super-geostrophic wind speed maximum. Such undisturbed inertial
297 oscillations are a widely known formation mechanism of nocturnal LLJ (NLLJ) (Sisterson and Frenzen,
298 1978). LLJs are often most pronounced during the early morning hours, typically before the onset of
299 daytime heating. During this time, the temperature inversion is typically strongest because nocturnal
300 cooling has been ongoing for several hours. After sunrise, the onset of daytime heating gradually disrupts
301 the stable boundary layer, reducing the occurrences of LLJ formation. Consequently, LLJs are less





frequent between noon and sunset.
In Figure 4(b), more than 70% of LLJs commonly occur at heights ranging from 0.3 km to 0.8 km
AGL in all seasons except summer. The vertical distribution of LLJs occurrences frequency in this study
also corresponds to previous studies. For example, Yan et al. (2021) found that 400 m AGL was the most
frequent height for the jet-nose appearing in the Huaihe River Basin. Wei et al. (2013) revealed that 76%
of the observed LLJs were found to occur at an average altitude below 600 m in the Yangtze River Delta
region. Following the classification of (Rife et al., 2010), the dominant type of LLJs in Hefei can be
identified as BLJs that occur mainly in the PBL below 1 km AGL. The highest occurrence frequency of
LLJs appeared between 0.5 km and 0.6 km AGL in all months other than July, with peak heights between
0.7 and 0.8 km AGL. The frequency of LLJs occurrences varies with months and heights in Hefei. LLJs
occurrences are most frequent during spring months, with decreasing frequency from March to May. Our
results are consistent with Yan et al. (2021), who found that LLJs were the most frequent in spring in
Huaihe River Basin based on long-term radiosonde observations from 2011 to 2017. The driving
mechanisms to LLJs include inertial oscillations under stable stratification, fronts and baroclinic weather
patterns in flat terrain, orographic and thermal effects in complex terrain. Considering the topography
and weather patterns, Hefei is prone to cyclones throughout the year, so the Asian monsoon system and
synoptic processes may be the most important influential factors in LLJs activities. In contrast, previous
studies on the LLJ climatology over other typical regions or cities showed different seasonal variations
of LLJs occurrences. For example, LLJs occur more often in spring and winter in Beijing while those
appear more frequently from October to December and from February to April in Guangzhou using long-
term wind profiler observations (Miao et al., 2018).

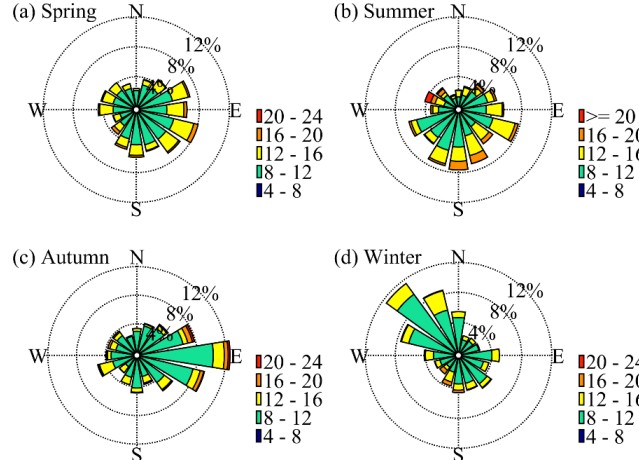

**Figure 5.** Wind rose charts for total LLJ events accounted for each season at Hefei. (a) Spring: Mar-May; (b) Summer:
Jun-Aug; (c) Autumn: Sep-Nov; and (d) Winter: Dec-Feb.
Figure 5(a) shows that the dominant wind directions of LLJs during spring are from the east (E) to
the southeast (SE). Furthermore, the maximum HWS of LLJs reaches up to 20 m s$^{-1}$ with more than half
of HWS exceeding 12 m s$^{-1}$. The varying dominant wind directions are associated with the transition
from the influence of the East Asian winter monsoon (EAWM) to its summer phase over Hefei. LLJs
occurrences peak in March due to its unique atmospheric conditions. During this time, the influence of
the winter monsoon is waning, but the full onset of the summer monsoon has not yet occurred. This





environment of a mix of cold and warm air masses is particularly favorable for LLJs formation.
Compared to March, LLJs occurrences are less frequent in April and May as the East Asian Summer
Monsoon (EASM) begins to take hold. As spring progresses, the strong baroclinic conditions that favors
LLJs formation begin to weaken. Because the temperature gradient between the cold north and the
warming south decreases driven by the growing influence of the western Pacific Subtropical High
(WPSH).

338       The overall occurrence frequency of LLJs during summer is lower than that in spring, but their
intensity is the strongest. In Figure 5(b), the predominant wind directions of LLJs are from the south (S)
and the east-southeast (ESE) with peak HWS reaching 20 m s$^{-1}$. During summer, the fully established
EASM is favorable for LLJs formation. Furthermore, the WPSH system extends northwestward from the
western Pacific Ocean towards eastern China, stabilizing the atmospheric conditions that favors LLJs
formation. The stronger the WPSH, the more intense the pressure gradient, which can lead to stronger
southeast-west winds at low levels. The LLJs occurrence generally peak in July, followed by June and
August. In July, the EASM is typically at the peak and the WPSH is usually at its most expansive and
positioned to exert the strongest influence over eastern China, including Hefei.

347       LLJs occurrences are less frequent during autumn and winter compared to spring and summer.
Figure 5(c) shows the predominant easterly wind direction (>12%) of LLJs throughout all autumn months,
with the maximum HWS reaching up to 24 m s$^{-1}$. As autumn approaches, the EASM transitions to the
EAWN and the WPSH further shifts eastward and southward (Figure A1). This shift exerts a weaker but
persistent influence that channels the air from the east. The least frequency of LLJs occurrences in winter
could be associated with general calm wind conditions in the lower troposphere (Figure 3d) and large-
scale synoptic systems, like cold fronts and high-pressure systems. These systems may not be conducive
to the formation of LLJs which typically require a specific set of atmospheric conditions, such as stable
conditions and wind shear. During winter, the predominant wind direction of LLJs during winter was
from the northwest (NE) in Figure 5(d), which is due to the dominance of the EAWN. The prevailing NE
wind of LLJs in winter was not as strong as in the other seasons, with maximum HWS reaching 16 m s$^{-}$
$^{1}$. Therefore, LLJs in Hefei are dominated by southwesterly winds in summer and northeasterly winds in
winter.

**3.4 Diurnal cycle of VWSH profiles for each season**

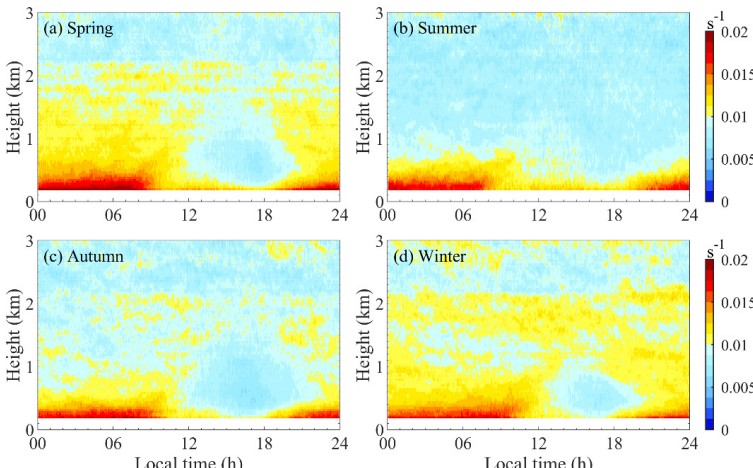




**Figure 6.** The same as in Fig. 3 but for VWSH.

363  VWSH depends directly on vertical wind profiles and exhibits both diurnal and seasonal variations
364 within the boundary layer, as shown in Figure 6(a)~(d). Due to surface friction, the wind speeds decrease
365 within the urban canopy, eventually reaching zero at ground level. These rapid changes in wind speed
366 create a large wind speed gradient, resulting in an increased shear intensity in the surface layer.
367 Throughout all seasons in Hefei, high VWSH values exceeding 0.015 m s$^{-1}$ per meter (hereafter denoted
368 by s$^{-1}$) are typically observed below 0.4 km.

369  Below 0.5 km, VWSH decreases from sunrise to the afternoon due to surface heating and increased
370 atmospheric mixing, which consequently led to a more uniform wind profile (Figure 3). In contrast, it
371 increases from sunset to early morning as surface cooling induces a temperature inversion, which creates
372 a stable boundary layer where winds aloft decouple from the surface. At night, a sharper wind speed
373 gradient with height is created under fully developed stable boundary layer, leading to maximum VWSH
374 in this layer. In the low to mid-level atmosphere (0.5~1 km), VWSH also varies diurnally, with relatively
375 lower values compared to VWSH below 0.5 km. Daytime VWSH in this layer is generally due to the
376 well-mixed boundary layer. But it can vary depending on local weather conditions and synoptic
377 influences. At night, high VWSH values above 0.01 s$^{-1}$ is usually associated with the presence of a LLJ
378 and/or a strong temperature inversion, with the maximum VWSH typically occurring just below the core
379 of the LLJ. In the upper level (> 1 km), VWSH is less influenced by the diurnal cycle and remained
380 relatively stable throughout the day. However, high VWSH can still occur in this layer when it is coupled
381 with LLJs or influenced by large-scale synoptic systems.

382  The seasonal variation of VWSH is closely linked to the region's climatic patterns, particularly the
383 influence of the East Asian monsoons, which drive significant changes in temperature, wind patterns,
384 and atmospheric stability throughout the year. In general, high VWSH values (> 0.015 s$^{-1}$) near the
385 surface are related to LLJs occurrences across all seasons. On the contrary, VWSH values above 1 km in
386 spring and winter are significantly larger compared to summer and autumn, which spatial pattern also
387 corresponds to vertical distributions of seasonal HWS profiles. During the two seasons, Hefei often
388 receives invasion of cold air/surge events, leading to strong winds. In winter, Hefei experiences strong
389 VWSH primarily due to the impact of the EAWM and large-scale synoptic systems, such as cold fronts
390 and jet-streams. Weaker solar heating in winter results in less pronounced diurnal variation of VWSH.
391 These synoptic systems also lead to significant VWSH (> 0.01 s$^{-1}$) above 1 km, which is characterized
392 by strong winds aloft. In spring, Hefei experiences strong VWSH due to the transitional atmospheric
393 conditions of the season. The diurnal variation shows a decrease in VWSH after 8:00 a.m. in the morning
394 compared to winter (Figure 6a). The variability in wind directions and speeds contributed to fluctuating
395 VWSH above 1 km influenced by shifting synoptic-scale systems and developing convective activity in
396 late spring.

397  In contrast, relatively lower VWSH values between 0.005 s$^{-1}$ and 0.01 s$^{-1}$ above 1 km are observed
398 in summer. The weather is dominated by the summer monsoon flow and localized convective systems.
399 These conditions result in a generally weaker VWSH with less pronounced diurnal variation compared
400 to other seasons. Due to significant vertical convective mixing, the wind profile becomes more uniform,
401 resulting in weaker VWSH above 1 km (Figure 6b). As the influence of the winter monsoon begins to
402 dominate, the strong winds aloft and weak surface winds contribute to an increasing VWSH in autumn
403 compared to summer (Figure 6c). Similar to winter, VWSH in autumn is more pronounced at night and
404 early morning due to the formation of temperature inversions. During the day, the reduction in VWSH
405 driven by vertical mixing is less noticeable than in summer, as the overall atmospheric stability increases.

### 3.5 Seasonal characteristics of the diurnal TKEDR profiles

As one of the characteristic features of the atmospheric turbulence, the TKEDR plays a crucial role in boundary layer parameterization schemes. It determines the rate at which turbulent kinetic energy is converted into thermal energy, directly influencing the vertical fluxes of momentum, heat, and mass. Long-term measurements of TKEDR will enhance our understanding of boundary layer dynamic processes and lead to more accurate simulations in atmospheric models.

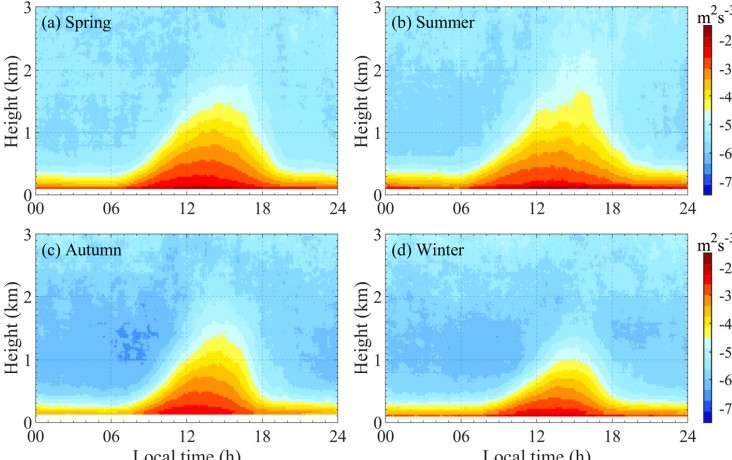

**Figure 7.** The same as Fig. 3 but for TKEDR.

Figure 7 illustrates the typical diurnal and seasonal cycles of the TKEDR profile. The TKEDR is highest near the surface, with typical values ranging from approximately $10^{-3}$ to $10^{-2}$ $m^2s^{-3}$, depending on the time of day and season. It decreases with height due to the diminishing influence of surface friction and thermal stratification. The convective boundary layer (or mixing layer) is clearly visible by noting where TKEDR is high. Diurnal variation starts from sunrise, as the increased temperature gradient between the surface and the above air enhances thermal buoyancy, which in turn promotes vertical convective mixing and turbulence. This causes TKEDR near the surface to grow and extend toward higher altitudes. In spring and summer, stronger and longer solar radiation leads to a more developed convective boundary layer, both in terms of duration and height, compared to autumn and winter. The convective boundary layer reaches its peak in the early afternoon, then begins to decay after 16:00 p.m., eventually returning to a shallow well-mixed layer near the ground, approximately 350 m in spring and summer, and around 250 m in autumn and winter. During the night, a stable atmospheric layer was formed near the surface and turbulence was primarily driven by mechanical factors (e.g., wind shear) rather than thermal convection. The complex urban surface roughness enhances wind friction, resulting in intensified turbulence, particularly during spring and summer when nocturnal LLJs occur more frequently. This increased turbulence contributes to the elevated TKEDR observed at night during these seasons.

As TKEDR decreases with altitude, its contour lines (though not explicitly plotted but evident from the color gradations in Figure 7) display a right-skewed shape, with a delayed peak time. This delay can be attributed to two factors: first, convective mixing activity takes time to propagate upward from the surface. Second, the ground cools more rapidly than the air in the late afternoon. Consequently, turbulence at higher altitudes lags low-level activity, reflecting the thermal-driven development of

turbulence and energy within the atmospheric boundary layer.
Here, we define the top of the convective boundary layer as the height where TKEDR reaches $10^{-4}$
$m^2s^{-3}$. It should be noted that this height can be different from the MLH given in the next section (Sect.
3.6), where the seasonal average MLH is calculated from the daily MLHs. We can see that the top of the
convective boundary layer during daytime in summer exhibits dramatic fluctuations, as shown in Figure
7(b), which cloud be attributed to the deep convective activities in the afternoon. Unstable atmospheric
stratification enhances vertical convection, leading to the formation of local convective clouds and
thunderstorms. These clouds reduce the amount of solar radiation reaching the surface, causing localized
cooling. Additionally, this process exacerbates the unevenness in the horizontal distribution of
temperature in the affected areas.
Overall, these seasonal and diurnal variations in TKEDR highlight the complex interactions
between surface properties, atmospheric stability, and weather systems in shaping the turbulence
characteristics within the boundary layer.

### 3.6 Seasonal variation of diurnal MLH for clear and cloudy days

The diurnal variations of MLH and BLH across different seasons in Hefei are depicted in Figure
8(a) and (b), respectively. The MLH is based on turbulence activities, while the BLH is based on the
vertical distribution of material (here aerosol). Therefore, both reflect the diurnal cycle of atmospheric
boundary layer dynamics, but there are some differences.

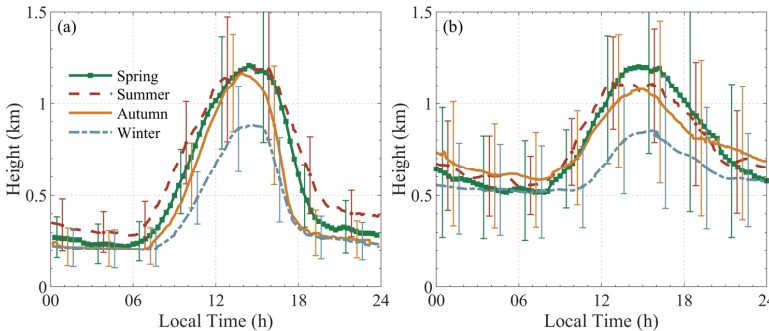

**Figure 8.** Time series plots of the seasonal average (a) MLH and (b) BLH at Hefei. The error bars represent one
standard deviation $\pm\sigma$, and their positions (corresponding to time) vary slightly in different seasons to facilitate
comparison.

After sunrise, surface heating induced by solar radiation promotes the devolvement of vertical
convective mixing, and drives the surface aerosols upward. When the temperature gradient between the
surface and air reaches its maximum, the MLH rises fastest, which appears at about 9:00-10:00 a.m. This
time varies with seasons, just as the sunrise time, with the earliest in summer, followed by spring, autumn,
and winter. The value of MLH at a certain time also shows the same seasonal relationship, except for the
afternoon in summer. Although solar radiation is highest at noon, the short-wave incident radiation
received by the surface in the afternoon is still greater than the long-wave outgoing radiation. Therefore,
the MLH continues to grow, reaching its maximum between 2:00 p.m. and 3:00 p.m. with about 1.2 km
in spring and summer, slightly lower in autumn, and 0.8 km in winter.
The similar afternoon peak of MLH between summer and spring could be attributed to several
factors. In the northern hemisphere, the summer solstice which occurs around June 21st or 22nd, is





relatively close to the spring period. This timing means that the transition from spring to summer is not always abrupt. Furthermore, high surface temperatures and increased evapotranspiration during summer lead to frequent convective clouds and precipitation. These factors reduce solar radiation received by the ground and weaken convective mixing, which can suppress the MLH. As a result, the seasonally averaged MLH reflects these cloudy conditions, leading to a peak height that may not be as high as one might expect on clear days.

In the late afternoon, as surface temperatures decrease due to radiative cooling, vertical convection weakens and turbulence kinetic energy dissipates more rapidly, leading to a faster decline in MLH compared to its increase in the morning. Meanwhile, the decrease in BLH is more gradual due to the slower rate of dry deposition of aerosols. It is noteworthy that the BLH curves exhibit larger fluctuations and significantly higher standard deviations compared to the MLH curves. This is primarily due to the considerable retrieval uncertainty in BLH measurements, which are influenced by aerosol distribution. Transboundary aerosols, clouds, and multilayer aerosols (e.g., residual layer) frequently affect these measurements, a well-recognized issue with aerosol-based BLH retrieval methods (Dang et al., 2019; Mei et al., 2022; Kotthaus et al., 2023).

During the night, the temperature inversion layer inhibits vertical thermal convection and mixing. Instead, mechanical mixing driven by wind shear becomes predominant, especially in the presence of low-level jets. Consequently, the MLH is typically highest in summer at about 0.3 km, followed by spring, and lowest in autumn and winter, about 0.2 km. In contrast, The BLH remains higher than the MLH, at approximately 0.5~0.7 km. The higher nocturnal BLH in autumn may be related to the transboundary transport of aerosols and meteorological factors. Both the MLH and the BLH continue to decrease and reach their minimum at sunrise in the next diurnal cycle.

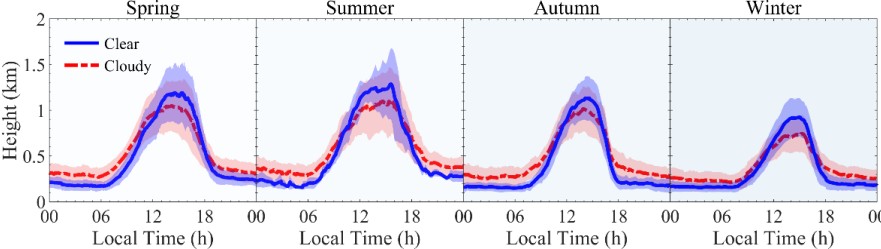

**Figure 9.** Time series plots of the seasonal average MLH (line) and one-sigma standard deviation (shaded area) for clear days and cloudy days during each season at Hefei, respectively.

To further investigate the influence of clouds on the development of MLH, we compared the seasonally averaged diurnal MLH under different weather conditions, as shown in Figure 9. The diurnal MLH showed significant differences between clear and cloudy days, and exhibited similar characteristics in each season. Overall, the diurnal variation of the MLH was less pronounced on cloudy days with a flatter curve, due to the modulation of clouds on the surface radiation budget. During daytime, the presence of clouds typically reduces surface heating by solar radiation, which inhibits the development of vertical convective mixing and results in a shallower mixed layer compared to clear weather conditions. The difference of MLH reaches its maximum of about 200 m in the afternoon. While during the night, clouds act as a "greenhouse" by absorbing longwave radiation from the ground and slowing down the radiative cooling, which results in a higher MLH compared to clear days. The mean difference in MLH between cloudy and clear days is about 100 m.





Note that, the diurnal MLH in summer showed relatively large variations, particularly on clear days.
This variability can be attributed to strong and variable convective activity, as well as to the limited
number of data samples. Plum rains and frequent convective clouds in summer lead to a much lower
proportion of sunny days than in other seasons.

## 4. Summary and Conclusions

In this study, three years of Doppler wind lidar measurements (spanning from June 2019 to June
2022) were utilized to characterize the PBL dynamics over Hefei City in western YRD, China. Compared
to aerosol lidars, the CDWL is capable of providing additional Doppler information including vertical
wind profiles, wind shear intensity, and turbulence mixing, with high spatiotemporal resolution.
Moreover, we identified LLJs events based on the nose characteristic of wind speed and retrieved both
the turbulence-based MLH and the aerosol-based BLH. Both seasonal and diurnal variations of these key
parameters were comprehensively analyzed to shed new insights into the structure and dynamics of the
PBL. The results are summarized as follows:
(1) Seasonal characteristics of wind profile: The frequency distribution of HWS exhibited a
rightward skew in all seasons, with lower values near the ground, and with a steady increase from 2 to 7
m s$^{-1}$ between 300 m and 3 km AGL, and a more rapid acceleration above 3 km. HWS% profiles in spring
and summer were more dispersed, with a lower frequency of high HWS occurrences (HWS > 10 m s$^{-1}$)
above 3 km. Seasonal HWD% profiles showed a predominance of westerly winds (270°±15°) above 3
km, while HWD within the PBL was more variable and chaotic. Seasonal VWS% profiles also exhibited
a right-skewed pattern with central values ranging between -0.2 m s$^{-1}$ and -0.1 m s$^{-1}$, indicating upward
motion. Winter, influenced by cold fronts associated with the winter monsoon, had the highest frequency
of negative VWS values, ranging from 4% to 7% below 3 km AGL.
(2) Diurnal characteristics of wind profile: A typical GWZ (HWS < 5 m s$^{-1}$) formed in the PBL
during the day in all seasons, with its diurnal variation strongly correlated with the development of the
mixing layer. The vertical height of high wind zone (> 8 m s$^{-1}$) during the day was much lower than at
night, particularly in winter, reaching 1.5 km between 11:00 a.m. and 4:00 p.m. In all seasons except
winter, a distinct local maximum in HWS between 0.4 km and 0.8 km was observed after 8:00 p.m. and
before 7:00 a.m. the next day. The phenomenon was most pronounced in summer due to the influence of
nocturnal LLJs.
(3) Monthly characteristics of LLJs: The dominant type was identified as BLJs in Hefei, with
occurrences being most frequent in spring (31.7%), followed by summer (24.7%), autumn (22.3%), and
winter (21.3%) in. LLJs were more frequently during the night and early morning throughout the year,
with 70% typically occurring at heights ranging from 0.3 km to 0.8 km AGL in all seasons except summer.
The highest occurrence frequency of LLJs appeared between 0.5 km and 0.6 km AGL in all months other
than July, with peak heights between 0.7 and 0.8 km AGL. Predominant wind directions of LLJs were
from the E and SE in spring, from S and ESE in summer, from E in autumn, and from NE in winter. LLJs
in summer were most intensified with largest frequency of high HWS (>16 m s$^{-1}$) and extended to
altitudes of up to 1.5 km.
(4) Seasonal and diurnal characteristics of VWSH, TKEDR, and BLH: High VWSH values
exceeding 0.015 s$^{-1}$ were typically observed below 0.4 km, which was usually associated with the LLJs
and/or strong temperature inversions at night. VWSH values above 1 km were significantly larger in
spring and winter compared to summer and autumn, correlating with vertical distributions of seasonal
HWS profiles. Strong wind speed gradients below and above the LLJs induced large vertical wind shear



intensity (up to 0.02 s$^{-1}$) and TKEDR (up to 10$^{-3}$ m$^2$s$^{-3}$) in the near-surface layer at night. TKEDR was generally highest near the surface, ranging from 10$^{-3}$ to 10$^{-2}$ m$^2$s$^{-3}$ in all seasons. The BLH exhibited larger fluctuations and greater standard deviations compared to the MLH. The peak MLH occurred between 2:00 p.m. and 3:00 p.m., reaching ~1.2 km in spring and summer, slightly lower in autumn, and around 0.8 km in winter. After sunset, it eventually returned to a shallow well-mixed layer near the ground (~350 m in spring and summer, and ~250 m in autumn and winter). Compared to clear days, cloud cover reduces the MLH by about 200 m at the afternoon peak time, while increasing it by approximately 100 m at night.

In conclusion, these analyses highlight the characteristics of PBL dynamics and their complex interactions with surface heating/cooling, atmospheric stability, and synoptic-scale weather patterns. The long-term statistical results will not only advance scientific understanding, but will also serve as essential references for formulating local standards and regional delineation, including vertical zoning, related to low-altitude economic activities, such as wind energy and drone logistics.

*Data Availability.* The Doppler wind lidar data used in this study can be provided for non-commercial research purposes upon request to the first author (Tianwen Wei: twwei@nuist.edu.cn ) . The ERA5 data sets are publicly available from the ECMWF website at https://cds.climate.copernicus.eu.

*Author contributions*. Tianwen Wei: Conceptualization, Methodology, Data curation, Formal analysis, Visualization, Writing – review & editing. Mengya Wang: Conceptualization, Writing – original draft, Methodology, Investigation. Kenan Wu: Resources, Data curation. Jinlong Yuan: Resources, Data curation. Haiyun Xia: Conceptualization, Supervision, Resources, Validation. Simone Lolli: Writing – review & editing, Validation.

*Conflict of Interest*. Some authors are members of the editorial board of Atmospheric Measurement Techniques.

*Financial support.* This work was supported by the National Natural Science Foundation of China (42405136), the Natural Science Research of Jiangsu Higher Education Institutions of China (23KJB170012).

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

# Appendix A

**Table A1.** Key Operating Parameters of the Doppler Lidar System

| Parameter | Parameter | Value |
|---|---|---|
| Laser | Wavelength (nm) | 1548 |
| | Pulse energy (μJ) | 300 |
| | Pulse duration (ns) | 600 |
| | Repetition rate (kHz) | 10 |
| | AOM frequency shift (MHz) | 80 |
| Telescope | Diameter (mm) | 100 |
| Data | Sampling frequency (MHz) | 250 |
| | Range gate length (m) | 30/60/150 |
| | Time resolution (s) | 1 |
| Scanning | Scanning mode | VAD |
| | Elevation angle (°) | 60 |
| | Azimuth angle (°) | 0-300 |


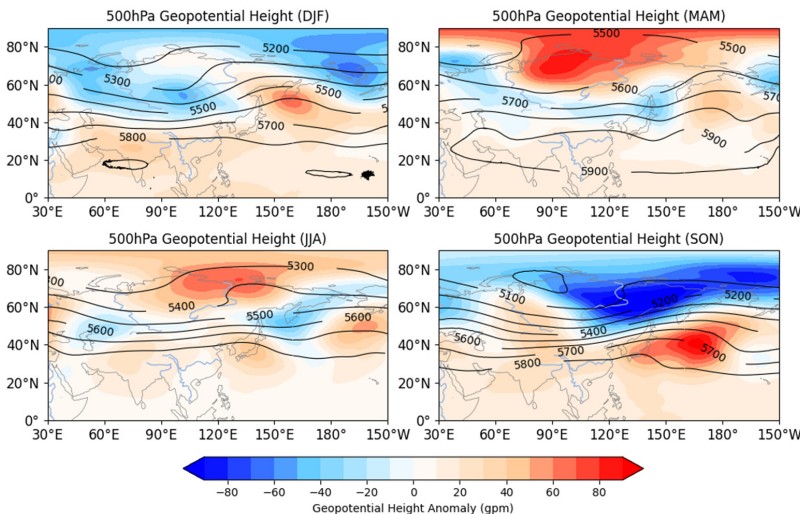

**Figure A1.** Seasonal distributions of 500-hPa geopotential height (contour, units: gpm) and geopotential height anomalies (shaded, units: gpm).