# Peer review of "Dynamics Using 3-Year Doppler Wind Lidar 3 Measurements in a Western Yangtze River Delta City, 4 China"

_Atmospheric Measurement Techniques, 2024_

## Author Comment (AC1)

* * *
Manuscript ID: AMT-2024-178
Title: Characterizing Urban Planetary Boundary Layer Dynamics Using 3-Year Doppler Wind Lidar Measurements in a Western Yangtze River Delta City, China
Authors: Tianwen Wei, Mengya Wang, Kenan Wu, Jinlong Yuan, Haiyun Xia, and Simone Lolli
* * *
Dear editor and reviewers

On behalf of the co-authors, thank you for giving us an opportunity to address the reviewers' concerns. We appreciate all the great efforts and constructive comments from the reviewers. We have revised the manuscript carefully according to the reviewer' comments and suggestions. Our point-by-point responses are appended below. All changes made in the revised manuscript are marked in blue. Attached please find the revised version of the manuscript, which we would like to submit for your kind consideration. We are looking forward to hearing from you!

Best regards!

Sincerely yours,

Tianwen Wei

School of Atmospheric Physics,

Nanjing University of Information Science & Technology

219 Ningliu rd. Nanjing, Jiangsu, CHINA, 210044.
* * *
**Anonymous Referee 1:**

This manuscript presents a comprehensive analysis of planetary boundary layer (PBL) dynamics over Hefei, China, based on 3-years of Doppler wind lidar measurements. The study is well-structured and highlights critical aspects of urban PBL, such as low-level jets (LLJs), mixing layer height (MLH), wind shear intensity and turbulence kinetic energy dissipation rates (TKEDR). The use of long-term lidar data provides valuable insights into both scientific understanding and practical applications, including atmospheric modeling and low-altitude economic activities. Overall, the study is well-conducted and contributes to the understanding of urban PBL dynamics. However, the manuscript could benefit from improvements in the depth of analysis and clarity of presentation, as outlined below.

We sincerely thank Reviewer 1 for constructive comments and suggestions, which have significantly improved the quality of our manuscript. Below, we provide our point-by-point responses to the comments:

Specific Comments:

1.  The key parameters of the lidar are currently provided in the Appendix. It is better to include them in Section 2.1 to improve accessibility for readers.

    Thanks for your suggestions. We have moved the key parameters of the Doppler wind lidar from the Appendix to Section 2.1, as suggested. This change improves the accessibility of the information for readers. The updated section now includes a table summarizing the key operating parameters of the lidar system.

2.  The manuscript references a wide range of studies, but the connections between previous research and the current work are not always explicitly stated. For example, in the discussion of LLJs, more emphasis on their implications for urban pollution or low-altitude economic activities in similar

East Asian climates would strengthen the relevance.

We appreciate the reviewer's insightful suggestion to strengthen the connection between previous research and our study, particularly in relation to the implications of low-level jets (LLJs) for urban pollution and low-altitude economic activities. In response, we have made the following revisions:

"The frequent occurrence of LLJs at heights below 1 km AGL enhances vertical mixing and turbulence within the lower atmosphere, breaking the decoupled boundary layer structure and restoring vertical heat, momentum, and pollutant exchanges. During nighttime, when stable stratification dominates, LLJs can reduce the accumulation of air pollutants near the surface by transporting them to higher altitudes. This mechanism is particularly important for urban areas like Hefei, where industrial and vehicular emissions often lead to air quality concerns. The temporal and vertical distribution of LLJs also has practical implications for low-altitude economic activities. For example, understanding LLJ dynamics provides valuable insights for designing safe and efficient drone flight routes, especially in areas with complex terrain or during nighttime operations. Additionally, the strong wind velocities associated with LLJs make them a key consideration for wind energy planning, particularly in optimizing the placement of wind turbines to maximize energy capture and efficiency."

3. Figure 5: The wind rose diagrams effectively summarize LLJ characteristics. Including a brief explanation of how these patterns relate to seasonal meteorological phenomena would enhance their interpretability.

Thank you for your valuable suggestions. We rewrite the description about Fig.5 and add a brief explanation of how these patterns relate to seasonal meteorological phenomena as the you suggested.

"Figure 5 shows the wind characteristics at the nose height of LLJs across different seasons. The dominant wind directions are southeast in spring, south in summer, and more concentrated in autumn (easterly) and winter (northwesterly). These characteristics in Hefei are closely related to the East Asian monsoon system and associated large-scale atmospheric circulations. In spring, LLJs occur most frequently due to the interaction between cold northerly air masses and warm, moist southerly flows during the transition from the East Asian Winter Monsoon (EAWM) to the East Asian Summer Monsoon (EASM). This dynamic interaction generates strong baroclinic conditions that are favorable for LLJ formation. In summer, the fully developed EASM and the northwestward expansion of the Western Pacific Subtropical High (WPSH) stabilize the boundary layer structure, leading to fewer LLJs compared to spring but with greater intensity (more than half of HWS exceeding 12 m s-1). The predominant wind directions during summer are southerly or southeasterly, reflecting the influence of the monsoonal flow. During autumn and winter, LLJs are less frequent as the WPSH retreats and the EAWM becomes dominant. Autumn marks the gradual transition, with occasional easterly LLJs influenced by the lingering WPSH. In winter, the stable conditions induced by the EAWM and associated high-pressure systems suppress LLJ formation, resulting in weak and infrequent northwesterly LLJs."

4. A figure of 500-hPa geopotential height is included in the Appendix but lacks sufficient explanation. If this figure is essential, consider elaborating on its implications for PBL dynamics and referencing related studies to support the discussion. Otherwise, it may be better to remove the

figure to streamline the manuscript.

Thank you for your valuable suggestions. We removed this figure and added two related references.

5. While the manuscript is generally well-written, certain sections and sentences could be simplified to improve readability. Additionally, minor errors should be corrected, such as:

   Line 78: "which the CBL" (Repeated phrase).

   Line 194: Missing space after "0.02 m s-1."

   Please review for consistency in terminology and formatting throughout the manuscript.

   Thank you for your valuable suggestions and for point out the errors in our manuscript. We have carefully reviewed the manuscript, simplified complex sentences to improve readability, and corrected the errors to ensure accuracy. Additionally, we have ensured consistency in terminology and formatting throughout the manuscript for improved clarity and coherence. The detailed corrections are highlighted in the red-line version of the revised manuscript.

**Other changes:**

**We added a new affiliation:**

"2China Meteorological Administration Xiong'an Atmospheric Boundary Layer Key Laboratory, Xiong'an New Area, Baoding 071800, China"

**We added a new Financial support:**

"the China Meteorological Administration Xiong'an Atmospheric Boundary Layer Key Laboratory (2023LABL-B11)"

**We added some related references:**

Barlow, J. F.: Progress in observing and modelling the urban boundary layer, Urban Clim., 10, 216–240, https://doi.org/10.1016/j.uclim.2014.03.011, 2014.

Barlow, J. F., Dunbar, T. M., Nemitz, E. G., Wood, C. R., Gallagher, M. W., Davies, F., O'Connor, E., and Harrison, R. M.: Boundary layer dynamics over London, UK, as observed using Doppler lidar during REPARTEE-II, Atmospheric Chem. Phys., 11, 2111–2125, https://doi.org/10.5194/acp-11-2111-2011, 2011.

Pérez-Ramírez, D., Whiteman, D. N., Veselovskii, I., Colarco, P., Korenski, M., and da Silva, A.: Retrievals of aerosol single scattering albedo by multiwavelength lidar measurements: Evaluations with NASA langley HSRL-2 during discover-AQ field campaigns, Remote Sens. Environ., 222, 144–164, https://doi.org/10.1016/j.rse.2018.12.022, 2019.

Pérez-Ramírez, D., Whiteman, D. N., Veselovskii, I., Ferrare, R., Titos, G., Granados-Muñoz, M. J., Sánchez-Hernández, G., and Navas-Guzmán, F.: Spatiotemporal changes in aerosol properties by hygroscopic growth and impacts on radiative forcing and heating rates during DISCOVER-AQ 2011, Atmospheric Chem. Phys., 21, 12021–12048, https://doi.org/10.5194/acp-21-12021-2021, 2021.

Wang, Y., Hu, H., Ren, X., Yang, X.-Q., and Mao, K.: Significant northward jump of the western pacific subtropical high: The interannual variability and mechanisms, J. Geophys. Res. Atmosphercs, 128, e2022JD037742, https://doi.org/10.1029/2022JD037742, 2023.

Wei, T., Wang, M., Jiang, P., Wu, K., Zhang, Z., Yuan, J., Xia, H., and Lolli, S.: Retrieving

aerosol backscatter coefficient using coherent doppler wind lidar, Opt. Express, 33, 6832–6849, https://doi.org/10.1364/OE.551730, 2025.

Yang, K., Cai, W., Huang, G., Hu, K., Ng, B., and Wang, G.: Increased variability of the western pacific subtropical high under greenhouse warming, Proc. Natl. Acad. Sci., 119, e2120335119, https://doi.org/10.1073/pnas.2120335119, 2022.

---

## Author Comment (AC2)

* * *
Manuscript ID: AMT-2024-178

Title: Characterizing Urban Planetary Boundary Layer Dynamics Using 3-Year Doppler Wind Lidar Measurements in a Western Yangtze River Delta City, China

Authors: Tianwen Wei, Mengya Wang, Kenan Wu, Jinlong Yuan, Haiyun Xia, and Simone Lolli
* * *
Dear editor and reviewers

On behalf of the co-authors, thank you for giving us an opportunity to address the reviewers' concerns. We appreciate all the great efforts and constructive comments from the reviewers. We have revised the manuscript carefully according to the reviewer' comments and suggestions. Our point-by-point responses are appended below. All changes made in the revised manuscript are marked in blue. Attached please find the revised version of the manuscript, which we would like to submit for your kind consideration. We are looking forward to hearing from you!

Best regards!

Sincerely yours,

Tianwen Wei

School of Atmospheric Physics,

Nanjing University of Information Science & Technology

219 Ningliu rd. Nanjing, Jiangsu, CHINA, 210044.
* * *
**Anonymous Referee 2:**

The paper deals with the analyses and exploitations of Doppler lidar measurements for a period of 3 years (2009-2022) in the city of Hefei, China. The focus is on the study of urban planetary boundary layer (PBL) deriving critical aspects such as low-level jets (LLJs), mixing layer height (MLH), wind shear intensity and turbulence kinetic energy dissipation rates (TKEDR). The study is well-structured, and I really appreciate that the analyses are supported by the data shown. I believe that the present study provides insight into PBL studies and is suitable for its publication in AMT.

We sincerely thank Reviewer 2 for constructive comments and suggestions, which have significantly improved the quality of our manuscript. Below, we provide our point-by-point responses to the comments:

I agree with the previous referee's comments and the analysis could be enriched. My suggestion is to add some specific study cases that serve to better illustrate the different situations that affect PBL and the rest of parameters.

Thank you for your valuable suggestion regarding the inclusion of case studies to enrich the analysis. While we acknowledge that detailed case studies can provide deeper insights into the dynamics of the planetary boundary layer under specific conditions, the primary focus of this study is to statistically characterize the diurnal and seasonal variations of key parameters, such as LLJs, MLH, and TKEDR. Instead, we have enhanced the analysis by expanding the discussions in the results section of the revised manuscript.

In future studies, we plan to conduct case-specific analyses under varying environmental conditions, incorporating aerosol observations and numerical simulations to further explore the interactions between the PBL dynamics and aerosol distributions. These efforts will complement the

findings of the current study and provide a more comprehensive understanding of boundary layer processes.

My only concern is that in the current state the conclusions section is only a summary, I miss the point out what are the novelties of the study, the limitations and the steps for the future.

Thank you for your valuable suggestion. We have rewrite the conclusions section to highlight the novelties, the limitations and the steps for the future study.

[revised manuscript text omitted]

Apart from that, I just detected some typos (i.e. legends axes are sometimes difficult to read, Figure captions can be improved).

Thank you for pointing out the typos in our manuscript. We have carefully reviewed the manuscript, improved the figure captions to provide more detailed explanations and ensure that the legends are clear and easy to read. Specifically, we have adjusted the color schemes in Figures 4 to improve readability. Additionally, we have ensured consistency in terminology and formatting throughout the manuscript for improved clarity and coherence.The detailed corrections are highlighted in the red-line version of the revised manuscript.

**Other changes:**

**We added a new affiliation:**
"2China Meteorological Administration Xiong'an Atmospheric Boundary Layer Key Laboratory, Xiong'an New Area, Baoding 071800, China"

**We added a new Financial support:**
"the China Meteorological Administration Xiong'an Atmospheric Boundary Layer Key Laboratory (2023LABL-B11)"

**We added some related references:**
Barlow, J. F.: Progress in observing and modelling the urban boundary layer, Urban Clim., 10, 216–240, https://doi.org/10.1016/j.uclim.2014.03.011, 2014.

Barlow, J. F., Dunbar, T. M., Nemitz, E. G., Wood, C. R., Gallagher, M. W., Davies, F., O'Connor, E., and Harrison, R. M.: Boundary layer dynamics over London, UK, as observed using Doppler lidar during REPARTEE-II, Atmospheric Chem. Phys., 11, 2111–2125, https://doi.org/10.5194/acp-11-2111-2011, 2011.

Pérez-Ramírez, D., Whiteman, D. N., Veselovskii, I., Colarco, P., Korenski, M., and da Silva, A.: Retrievals of aerosol single scattering albedo by multiwavelength lidar measurements: Evaluations with NASA langley HSRL-2 during discover-AQ field campaigns, Remote Sens. Environ., 222, 144–164, https://doi.org/10.1016/j.rse.2018.12.022, 2019.

Pérez-Ramírez, D., Whiteman, D. N., Veselovskii, I., Ferrare, R., Titos, G., Granados-Muñoz, M. J., Sánchez-Hernández, G., and Navas-Guzmán, F.: Spatiotemporal changes in aerosol properties by

hygroscopic growth and impacts on radiative forcing and heating rates during DISCOVER-AQ 2011, Atmospheric Chem. Phys., 21, 12021–12048, https://doi.org/10.5194/acp-21-12021-2021, 2021.

Wang, Y., Hu, H., Ren, X., Yang, X.-Q., and Mao, K.: Significant northward jump of the western pacific subtropical high: The interannual variability and mechanisms, J. Geophys. Res. Atmospheres, 128, e2022JD037742, https://doi.org/10.1029/2022JD037742, 2023.

Wei, T., Wang, M., Jiang, P., Wu, K., Zhang, Z., Yuan, J., Xia, H., and Lolli, S.: Retrieving aerosol backscatter coefficient using coherent doppler wind lidar, Opt. Express, 33, 6832–6849, https://doi.org/10.1364/OE.551730, 2025.

Yang, K., Cai, W., Huang, G., Hu, K., Ng, B., and Wang, G.: Increased variability of the western pacific subtropical high under greenhouse warming, Proc. Natl. Acad. Sci., 119, e2120335119, https://doi.org/10.1073/pnas.2120335119, 2022.